# Utilisation of Colorectal Cancer Screening Tests in European Countries by Type of Screening Offer: Results from the European Health Interview Survey

**DOI:** 10.3390/cancers12061409

**Published:** 2020-05-29

**Authors:** Rafael Cardoso, Feng Guo, Thomas Heisser, Michael Hoffmeister, Hermann Brenner

**Affiliations:** 1Division of Preventive Oncology, German Cancer Research Center (DKFZ) and National Center for Tumor Diseases (NCT), 69120 Heidelberg, Germany; h.brenner@dkfz.de; 2Medical Faculty Heidelberg, University of Heidelberg, 69120 Heidelberg, Germany; f.guo@dkfz.de (F.G.); t.heisser@dkfz.de (T.H.); 3Division of Clinical Epidemiology and Aging Research, German Cancer Research Center (DKFZ), 69120 Heidelberg, Germany; m.hoffmeister@dkfz.de; 4German Cancer Consortium (DKTK), German Cancer Research Center (DKFZ), 69120 Heidelberg, Germany

**Keywords:** colorectal cancer, screening, faecal tests, colonoscopy, utilisation, Europe

## Abstract

In the past two decades, an extensive rollout of colorectal cancer (CRC) screening programmes has been initiated in European countries with a large heterogeneity of screening offers. Using data from a population-based cross-sectional survey conducted between 2013 and 2016 in all European Union countries, we analysed the utilisation of faecal tests and colonoscopy among people aged 50–74 years and the factors associated with uptake by type of screening offer. We observed the highest utilisation of either test for countries with fully rolled out organised programmes with faecal tests (ranging from 29.7% in Croatia to 66.7% in the UK) and countries offering both faecal tests and colonoscopy (from 22.7% in Greece to 70.9% in Germany). Utilisation was very low for countries with no programme (from 6.3% in Romania to 30.5% in Norway). Younger age (50–54 years), longer time since last consultation with a doctor and a lifestyle score associated with increased CRC risk were significantly associated with lower test use, a pattern observed across all types of screening offers. Our results suggest that more countries should implement organised programmes with faecal immunochemical tests, in combination with alternative endoscopy offers where resources allow. Furthermore, there is a large potential for increasing screening use in Europe by better reaching the younger eligible individuals, those who have not been to the doctor recently and those at increased risk for CRC.

## 1. Introduction

Colorectal cancer (CRC) was estimated to be the third most common cancer and the second leading cause of cancer death worldwide in 2018 [1]. With age-standardised incidence and mortality rates of 45.9 and 18.5 (per 100,000), respectively, CRC poses a particularly high burden for the European Union (EU) countries [2]. However, due to the slow development of the disease and the existence of effective screening strategies, a large proportion of CRC cases and related deaths could be either prevented or detected early and successfully treated [3,4,5]. Specifically, faecal tests [6,7,8] and lower gastrointestinal endoscopy (colonoscopy and flexible sigmoidoscopy) [4] have been shown to considerably reduce CRC incidence and mortality. Faecal tests are non-invasive tests that detect occult blood in the stool by targeting either haem (guaiac-based faecal occult blood test, gFOBT) or human haemoglobin (faecal immunochemical test, FIT); lower gastrointestinal endoscopy consists in the examination of the entire large bowel (colonoscopy) or its distal part (flexible sigmoidoscopy) by an endoscope that is also able to remove precancerous lesions. Other tests, namely stool DNA tests, have been recommended as alternative methods by several US expert groups and societies [9,10,11]; however, no European country has considered them in their screening programmes to date [12,13].

Already in 2003, recognising the effectiveness of CRC screening, the EU Council called upon its Member States to implement organised CRC screening with faecal tests targeting the population aged 50–74 years [14]. Some EU countries have meanwhile followed the recommendations and implemented either regional or nationwide organised programmes, while others have been offering CRC screening mainly in an opportunistic manner (on an ad hoc basis, essentially faecal tests and/or colonoscopy as primary screening modalities) or no CRC screening [12]. These differences in screening offer are expected to be reflected in different utilisation rates and patterns across different demographic, socioeconomic and at-risk groups in the EU countries.

Between 2013 and 2016, socioeconomic and health-related factors, including the use of faecal tests and colonoscopy, were ascertained in all EU Member States, Iceland, Norway and the UK through the second wave of the European Health Interview Survey (EHIS) [15]. We first reviewed relevant characteristics of CRC screening offers in the EHIS-participating countries and subsequently, using the EHIS, analysed the use of faecal tests and colonoscopy, as well as factors associated with uptake by type of screening offer.

## 2. Results

### 2.1. CRC Screening Offers in the EU Countries, Iceland, Norway and the UK

A summary of relevant characteristics of CRC screening programmes in the EHIS-participating countries is presented in Table 1. The timing of enrolment, the type of programmes, the screening tests used and the target age groups vary considerably across countries. By the time the EHIS was conducted, four countries had fully rolled out their organised programmes with faecal tests (Croatia, France, Slovenia and the UK) and eleven were in the process of rolling out their organised programmes or only offered screening in certain regions (Belgium, the Czech Republic, Denmark, Finland, Ireland, Italy, Malta, Lithuania, the Netherlands, Spain and Sweden). Seven countries also offered faecal tests but predominantly in an opportunistic manner (Austria, Germany, Latvia, Luxembourg, Lithuania, Portugal and Slovakia). Furthermore, colonoscopy was offered as a primary screening modality in eight countries (Austria, the Czech Republic, Germany, Greece, Iceland, Luxembourg, Portugal and Slovakia). The remaining countries did not have any screening programme in place or small-scale pilot programmes only.

### 2.2. Utilisation Rates by Type of CRC Screening Offer 

Among the four countries where organised programmes had been rolled out nationally, three were found to have faecal test utilisation rates higher than 50% (UK: 59%, Slovenia: 56%, France: 51%); the exception was Croatia with only 22% (Figure 1). When looking at countries where organised programmes were being rolled out at the time of the survey, large variations in faecal test use were observed, ranging from 10% in the Netherlands to 42% in the Czech Republic. As for countries where faecal tests were offered mainly through opportunistic approaches, the lowest utilisation rate was observed for Greece (11%) and the highest rates for Germany (51%) and Austria (49%), where utilisation levels were close to those reported for the UK, Slovenia and France. Faecal test use was very low (below 15%) for all countries/age groups where no programme was in place. A proportion of older adults, no longer targeted by screening programmes, ranging from 8% in Spain to 30% in Italy, reported to still be up-to-date with faecal tests.

As far as colonoscopy use is concerned, utilisation levels below 32% were observed for all countries where colonoscopy was not offered as an alternative primary screening modality and below 20% for the majority of countries where no screening programme was available (Figure 2). Among countries offering colonoscopy as an alternative primary screening test, utilisation levels were highest in Austria (52%), Germany (51%) and Luxembourg (49%) and lowest in Greece (15%) and Slovakia (15%). 

When looking at the utilisation of either faecal tests or colonoscopy, we observed the highest utilisation rates for countries where faecal tests were offered within an organised programme fully rolled out nationally and for countries with offers of both faecal tests and colonoscopy (Figure 3). Among the former, the UK was found to have the highest utilisation rates (67%) and Croatia the lowest (30%); among the latter, the highest utilisation rates were observed for Germany (71%) and the lowest for Greece (23%). Utilisation was very low (< 31%) for all countries with no programme.

### 2.3. Determinants of Test Use by Type of CRC Screening Offer

From all the potential predictors studied, age, health care use and the lifestyle score were found to be the strongest predictors of test use (Appendix A). Specifically, when compared to those aged 60–64 years, individuals aged 50–54 years were much less likely to have undergone either faecal tests or colonoscopy (ORs ranging from 0.57 (95% CI: 0.50 to 0.64) among countries offering colonoscopy as an alternative primary screening modality to 0.77 (95% CI: 0.64 to 0.92) among countries with no programme in place). Furthermore, those who reported not having had a consultation with a physician within the previous 12 months were overall 40%–60% less likely to have undergone any test (ORs ranging from 0.44 (95% CI: 0.37 to 0.53) to 0.62 (95% CI: 0.53 to 0.73)), a pattern observed across all types of screening offers. A similar pattern was observed for people with lower lifestyle scores (especially those with scores 0 or 1) who had 22%–36% lower likelihood of having undergone either test (ORs ranging from 0.64 (95% CI: 0.48 to 0.86) to 0.78 (95% CI: 0.67 to 0.92)).

Besides age, health care use and the lifestyle score, self-perceived health was found to strongly predict colonoscopy use. In particular, when compared to people who perceived their health as good or very good, those who reported a less than good health status were found to be more likely to have undergone colonoscopy within the previous 10 years. This association was especially pronounced in countries with national coverage of organised programmes (OR: 1.58, 95% CI: 1.34 to 1.86) and in countries with no programme in place (OR: 1.70, 95% CI: 1.55 to 1.87). 

## 3. Discussion

### 3.1. Principal Findings

This study provides an overview on the use of faecal tests and colonoscopy among the general average-risk population aged 50–74 years in the EU countries, Iceland, Norway and the UK by the type of CRC screening offer. Large differences in utilisation were observed between the different countries, with only five (Germany, Austria, the UK, Slovenia and France) reaching proportions of the eligible population up-to-date with either test higher than 60%. Overall, countries with nationwide coverage of organised programmes with faecal tests and countries which provided both faecal tests and colonoscopy as alternative screening options were found to have comparable, and the highest, proportions of the population up-to-date with either test. Utilisation was much lower or almost nonexistent in countries where no screening programme was in place. Of the various potential predictors investigated, being in the youngest age range eligible for screening, the absence of a recent consultation with a physician and being at higher risk for CRC based on lifestyle characteristics were strongly associated with the decreased use of faecal tests or colonoscopy, irrespective of the type of CRC screening offer.

### 3.2. Strengths and Weaknesses of the Study

The EHIS constitutes the reference source of evidence to support health-related policy making in the EU region. A major strength is that all countries had to follow detailed rules and recommendations for data collection in order to ensure high comparability levels and to use nationally representative probability samples [60]. Furthermore, to our knowledge, no international comparison on the use of faecal tests and colonoscopy has been carried out across all EU countries by the type of CRC screening and using nationwide population-based data [61]. The utilisation rates reported in this study will be of great importance when it comes to analysing and comparing the potential effects of screening on CRC incidence and mortality in the EU countries. These data may also be used in future studies as model input to project CRC incidence and mortality and to potentially improve screening offers.

The study also has some limitations that should be considered when interpreting its findings. First, the data were mostly collected from self-reports which may have led to reporting and recall biases and therefore the over- or underreporting of test use and the explanatory factors included in the analysis [62]. Second, given that the countries which relied on self-administered data collection had overall higher non-response rates, selection bias may have been introduced. Third, the data did not allow for discrimination between screening and diagnostic colonoscopies. Fourth, the cut-offs used to dichotomise the individual items of the lifestyle score, especially smoking and alcohol consumption, were slightly different from the ones outlined in the recommendations, and no information on diet was available [63]. This may have led to the attribution of a less appropriate score to a small proportion of respondents. Lastly, some countries offer other screening modalities that we were not able to analyse. Specifically, in England and Italy (Piedmont and Veneto regions), flexible sigmoidoscopy is also offered as an alternative screening test [38,64]. Therefore, especially for the UK, the proportion of individuals up-to-date with CRC screening is likely to be higher than the one based on faecal tests and colonoscopy only. 

### 3.3. Comparison with Other Studies and Interpretation of Results

Comparisons across European countries were first performed in the context of the Survey of Health, Aging and Retirement in Europe (2004/2005) when most countries had not yet implemented CRC screening programmes [65]. Apart from Austria and Germany, where faecal tests and colonoscopy had already been offered in an opportunistic manner, utilisation rates were very low for all countries (below 25%). Similar utilisation rates were now found for the group of countries which still do not have CRC screening programmes.

More recently, the status [13], coverage [16] and performance [41] of CRC screening have been disclosed for organised programmes implemented in some EU countries in the context of the second European screening report (data collection from 2011 to 2014) [20]. For countries with fully implemented organised programmes nationally, whose estimates of faecal test use can be compared to the ones from this report, similar results were observed for Slovenia and the UK (utilisation rates about 50%–60%). For France, we found considerably higher utilisation rates (51% compared to < 30% reported by Senore et al.) [41] which may be in part explained by the discrepancy between the more recent data collection for EHIS (2013–2016) and the screening report (2012). The much lower utilisation rates observed for Croatia are in line with those described in national reports and have been partly attributed to a lack of CRC screening awareness in the population [21,66]. Other issues that have been raised relate to the absence of colonoscopy facilities on the Croatian islands, which make it difficult to follow up on a positive faecal test [21].

As for countries where organised programmes were still being rolled out, or where not all regions were covered, our national estimates reflect such gradual implementations. For instance, in Spain and Sweden CRC screening had only been implemented in some regions, and in the Netherlands the EHIS data collection took place in the year the programme was launched. Therefore, low utilisation rates would be expected. Nonetheless, the high participation rates that have been reported for these countries among those invited to screening [41] suggest that similar utilisation rates to the ones observed for France, Slovenia and the UK are likely to be achieved once the programmes are fully implemented nationally. On the other hand, the comparatively high utilisation of faecal tests observed for the Czech Republic may be explained to a large extent by the existence of a parallel opportunistic programme already adopted in 2000. For other countries with higher utilisation rates (e.g., Denmark, Ireland and Italy), the EHIS was implemented when the nationwide screening programmes had already been rolled out for at least one year.

Large variations in test use were also observed across countries providing CRC screening mainly in an opportunistic manner and seem to be highly related to differences in health care systems and available resources. Specifically, a lack of prioritisation of cancer prevention and low involvement of physicians in early detection may help explain the low proportions of the population up-to-date with either test observed for countries such as Greece and Slovakia [67,68,69,70]. In countries with much higher proportions, namely Austria and Germany, CRC screening has been in place for decades and awareness of CRC screening is high [12,17,71]. The high proportions observed for the latter additionally suggest that, for countries where the necessary resources are available, offering colonoscopy as an alternative option may contribute towards achieving high CRC screening utilisation rates. This is further supported by a recent study from Germany which has found an increase in CRC screening uptake rates after the introduction of screening colonoscopy as an alternative to faecal tests [32].

Looking at the utilisation among countries where no programme or only a small-scale programme was in place, the very low levels observed for all countries may mainly reflect test use for other purposes. In particular, a large proportion of the population up-to-date with colonoscopy is likely to have undergone colonoscopy for the clearance of symptoms rather than for screening purposes. This may also explain, to a large extent, the considerably higher odds of having undergone colonoscopy among people who reported a less than good health status in comparison to those who perceived their health as good or very good. Furthermore, the lower likelihood of being up-to-date with faecal tests or colonoscopy for all types of screening offers among the youngest age groups eligible for screening, and those at the highest risk for CRC, suggests a large potential for reducing the high CRC burden in Europe by efforts to better reach these people by screening offers. General practitioners could play a major role in this context, as supported by the strong association that was consistently seen between having seen a doctor in the last year and the use of CRC screening offers even within organised programmes, in which the eligible population is informed with an invitation letter and the faecal test is sent along. 

Given the very recent implementation of screening in most countries, its effects on CRC mortality are expected to be fully disclosed only in the next decades. However, the reported variations in the timing of implementation, screening offers and uptake across countries suggest that large differences in the progress towards CRC mortality reduction will be observed across EU countries.

## 4. Materials and Methods

### 4.1. Study Design and Study Population

This study used data from the second wave of the EHIS. EHIS is a cross-sectional survey aimed at providing harmonised and highly comparable data across EU countries to support health policies and address health inequalities and social exclusion [72]. The survey was conducted in 30 countries (all EU countries, Iceland, Norway and the UK) under the Commission Regulation (EU) No 141/2013 between 2013 and 2016 and targeted non-institutionalised individuals aged 15 years and older residing in these countries at the time of data collection. 

In order to ensure national representativeness and comparability of the data, each country used probability sampling techniques and followed methodological recommendations set out by Eurostat [60]. A median response rate of about 60% was achieved across countries, ranging from less than 50% in Austria, Denmark, Finland, Germany and Luxembourg to over 90% in Cyprus and Portugal. Potential non-response bias was addressed by calculating and attributing to each respondent a weighting factor that ensures each country is considered in proportion to its demographic distribution. Further details can be found in the Eurostat quality report [72]. 

For this analysis, only individuals aged 50–74 years, who are commonly considered eligible for CRC screening, were included [12,13]. Moreover, only the participants whose survey responses were provided by themselves were included; proxy interviews were excluded to rule out potential, and very likely, inaccurate responses. In total, 128,496 individuals aged 50–74 years participated in the study. Among them, 3121 were drawn from proxy interviews and another 3479, 2419 and 4210 did not provide data on the utilisation of faecal tests, colonoscopy and either test, respectively. Hence, 121,896 respondents were included for analyses of faecal test use, 122,956 for colonoscopy use and 121,165 for either faecal test or colonoscopy use (Figure 4).

### 4.2. Data Collection

The data collection period varied from 3 to 21 months across the different countries. Various data collection methods were adopted, consisting of face-to-face interviews, telephone interviews, post and a combination of different methods. The survey was translated from English into the local languages of each individual country and pre- or pilot-tested in most countries [72]. 

### 4.3. Measures

The outcome measures of faecal test use (either gFOBT or FIT) within the previous 2 years, colonoscopy within the previous 10 years and the use of either test in the respective time frames were ascertained by enquiring the respondents about the last time these tests were undertaken. These time frames are the most widely used screening intervals for the average-risk population in the EHIS-participating countries [12,13]. The explanatory variables tested as potential determinants of test use encompassed demographic and socioeconomic factors (sex, age, location of residence, marital status, education and household income), health care use (last time of a consultation with a general practitioner and a medical or surgical specialist) and health-related factors (self-perceived health and a healthy lifestyle score). 

### 4.4. Derivation of the Healthy Lifestyle Score

A healthy lifestyle score was created based on the established evidence on CRC risk and protective factors, notably smoking, alcohol consumption, physical activity and body mass index (BMI) [63] (Appendix A). It was adapted from a score created by Carr et al. that was associated with lower risk for all stages of CRC [73,74]. Specifically, respondents were attributed one point for the following low-CRC-risk behaviours: non-smoking or occasional smoking, consumption of less than two alcoholic drinks per day, being physically active (at least 150 min per week spent on doing sports, fitness or recreational physical activities, in line with World Health Organization Global Recommendations on Physical Activity for health [75]) and having a BMI below 25 kg/m^2^, i.e., not being overweight or obese [63]. 

Data on physical activity were not available for Belgium and on alcohol consumption for France and Italy. Therefore, a score ranging from 0 to 3, instead of 0 to 4, was attributed to each respondent from these three countries. For the Netherlands, data on both alcohol consumption and physical activity were not available, thus it was not considered in the multivariate analyses for which the lifestyle score was used as a covariate.

### 4.5. Classification of Countries/Age Groups by Type of CRC Screening Offer

A review of the relevant characteristics of CRC screening programmes in the 30 EHIS-participating countries was carried out. Information was retrieved from articles on the second European screening report [20], websites from national governments and cancer registries, as well as from a literature search in PubMed using the keywords: colorectal cancer screening, colonoscopy, guaiac faecal occult blood test and faecal immunochemical test, alongside the term “Europe” and each country name [12,13,16,17,18,19,20,21,22,23,24,25,26,27,28,29,30,31,32,33,34,35,36,37,38,39,40,41,42,43,44,45,46,47,48,49,50,51,52,53,54,55,56,57,58,59]. The characteristics that we reviewed and summarised served as a basis for the analyses of EHIS data.

For the analyses of faecal test use, the following five categories were created: (A) Nationwide fully rolled out organised programmes with faecal tests; (B) Organised programmes with faecal tests in partial rollout or with regional coverage only at the time of EHIS data collection; (C) Opportunistic programmes with faecal tests; (D) No programme with faecal tests or a small-scale pilot programme only; (E) Other, that is, age groups not targeted by screening programmes but to whom faecal tests may have been offered within the preceding two years when these individuals were eligible for screening.

To analyse colonoscopy use or the use of either faecal tests or colonoscopy, the abovementioned groups (A) and (B) were kept and two additional groups were created, namely: (C) Colonoscopy offered as an alternative primary screening modality; (D) No programme, small-scale organised programme, or opportunistic programme with faecal tests as the first-line method only. Unlike for faecal test use, an additional category (E) with older age groups not targeted by the screening programmes was not created, as these older birth cohorts had within the past 10 years been among the age groups eligible for colonoscopy (either as a primary screening modality or as a follow-up test). These were therefore included together with the younger age groups targeted by screening programmes in their respective category. 

Appendix A provides an overview of the countries and age groups included in the different categories created to analyse the utilisation of faecal tests, colonoscopy and either test. These categories reflect the status of CRC screening implementation at the time the EHIS was carried out.

### 4.6. Statistical Analyses

Prevalence estimates of faecal test use within the previous 2 years, colonoscopy use within the previous 10 years and the use of either of the two tests were determined for each country according to the abovementioned categories. In order to explore the potential determinants of test use, multivariate logistic regression models were employed and odds ratios (ORs) and confidence intervals (CIs) were obtained. The log (OR) and their standard errors were subsequently computed for each country and subgroup meta-analyses were performed to estimate ORs and CIs by category/type of screening offer using the metagen function in RStudio Version 1.2.1335 (RStudio, Inc., Boston, MA, USA). The estimates obtained from the random-effects subgroup meta-analyses were extracted and summarised.

For both the prevalence estimates and odds ratios, individual weights were applied, accounting for the units’ probability of selection, non-response, over- or under-representation of certain population groups and calibrated with country-specific distributions of the population with regards to demographic characteristics [60]. Variances were calculated using the Taylor series linearisation method that takes into account the complexity of the survey design.

Except for the meta-analyses, all other analyses were conducted using SAS 9.4 (SAS Institute Inc., Cary, NC, USA).

### 4.7. Ethics Approval

The EHIS was carried out according to the Regulation (EC) No 1338/2008 of the European Parliament and of the Council of 16th December 2008. The data collected by each country have been specified in the Commission Regulation (EU) No 141/2013. Ethics approval was obtained on a national level by the institutions responsible for the survey implementation. Further information can be found in the quality report released by Eurostat: https://ec.europa.eu/eurostat/documents/7870049/8920155/KS-FT-18-003-EN-N.pdf/eb85522d-bd6d-460d-b830-4b2b49ac9b03. 

## 5. Conclusions

The proportion of the population up-to-date with faecal tests or colonoscopy had remained low for the majority of the EU countries in 2013–2016. Nevertheless, in line with previous evidence, our study suggests that, when completely rolled out, organised programmes with faecal tests have the potential to reach a high proportion of the target population. In programmes with faecal tests, FITs rather than gFOBT, should be offered, given their superior diagnostic performance [76,77]. Moreover, offering the choice to undergo either faecal tests or colonoscopy was shown to attain similar proportions of the population up-to-date with CRC screening even if the offer was provided in an opportunistic manner only. Therefore, in the EU context, and based on the resources available in each country, it seems plausible that organised programmes offering faecal tests and colonoscopy as alternative test options might lead to even higher CRC screening uptake levels than organised programmes solely offering faecal tests. It also seems plausible that even higher uptake rates could be achieved by complementary offers of faecal tests and flexible sigmoidoscopy as an almost equally effective, but less invasive, endoscopy alternative [4,78]. Our results furthermore underline the important role physicians could have in motivating people to use available CRC screening offers, in particular the younger age groups eligible for screening and those at increased risk for CRC.

## Figures and Tables

**Figure 1 cancers-12-01409-f001:**
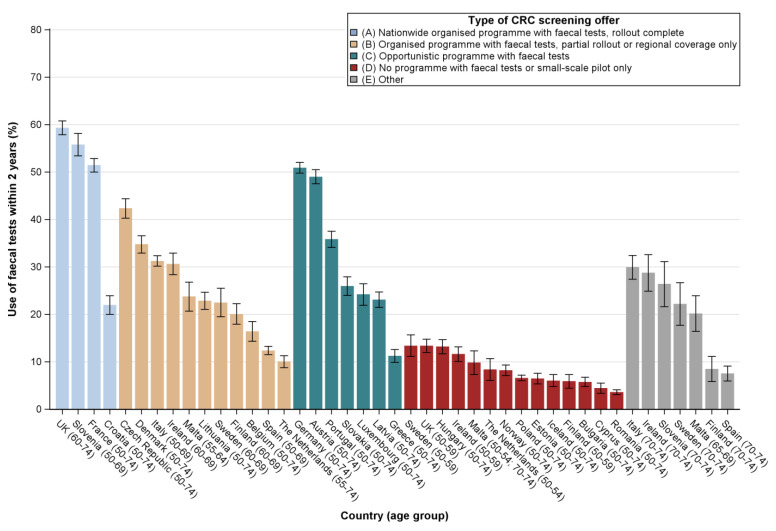
Prevalence estimates of faecal test use within the previous 2 years among the general population aged 50–74 years in all EU countries, Iceland, Norway and the UK, by type of screening offer (data: EHIS, 2013-2016).

**Figure 2 cancers-12-01409-f002:**
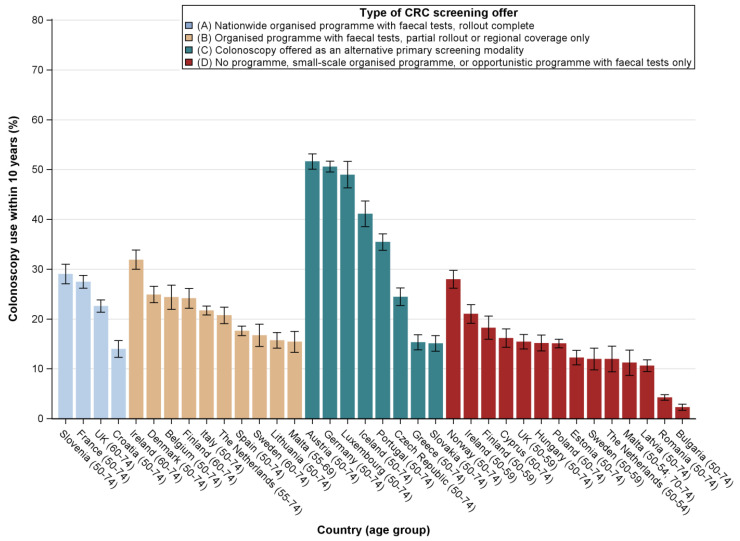
Prevalence estimates of colonoscopy use within the previous 10 years among the general population aged 50–74 years in all EU countries, Iceland, Norway and the UK, by type of screening offer (data: EHIS, 2013–2016). For Sweden, colonoscopy use was determined within the past 5 years.

**Figure 3 cancers-12-01409-f003:**
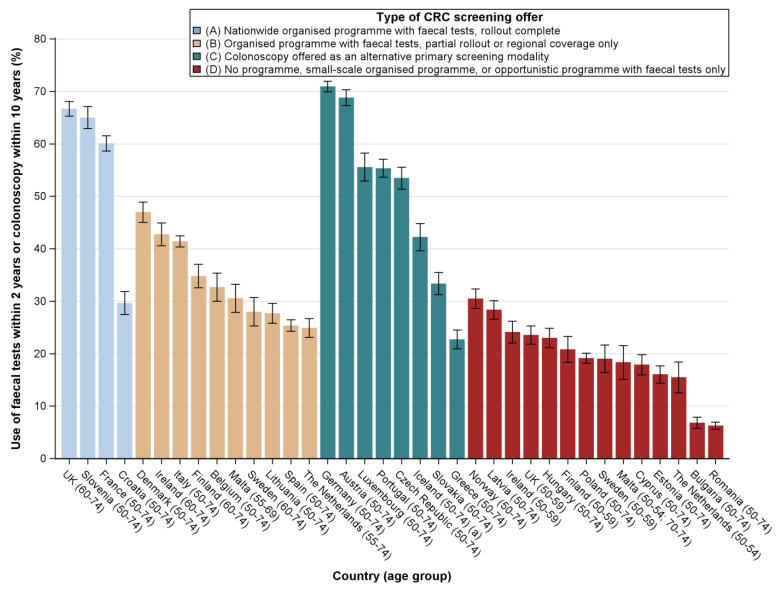
Prevalence estimates of faecal test use within the previous 2 years or colonoscopy use within the previous 10 years among the general population aged 50–74 years in all EU countries, Iceland, Norway and the UK, by type of screening offer (data: EHIS, 2013–2016). For Sweden, colonoscopy use was determined within the past 5 years.

**Figure 4 cancers-12-01409-f004:**
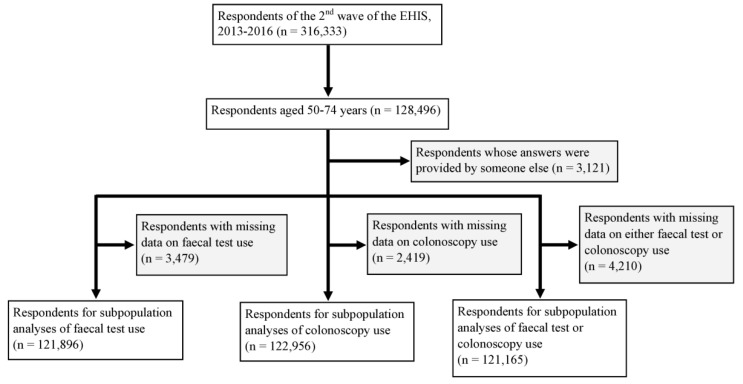
Flowchart of respondents included in, and excluded from, the analyses due to a lack of data on the outcome measures or proxy interviews.

**Table 1 cancers-12-01409-t001:** Characteristics of colorectal cancer screening programmes in the EU countries, Iceland, Norway and the UK ^a^.

Country	Type of Programme	Are the Tests Paid for?	Screening Test ^a^	Year of Programme Initiation and, If Applicable, Termination	Age Groups (years)	Screening Interval (years)	Is the Faecal Test Mailed?	Invitation Coverage in 2013 in the 50–75 Age Group (%) ^b^	EHIS Data Collection (mm/yyyy)	References
**Austria**	Opportunistic	Yes	gFOBT	1980	40+	1	NA^c^	NA	09/2013–06/2015	[12,13,16,17]
	Opportunistic	Yes	Colonoscopy	2005	50+	7–10	NA	NA		
**Austria** (state of Burgenland)	Organised	Yes	FIT	2003	40–80	1	Yes	No data		
**Belgium** (Wallonia/Brussels)	Organised	Yes	gFOBT	2009–2016	50–74	2	No	99.0	01/2013–12/2013	[12,13,16,18,19]
			FIT	2016						
**Belgium** (Flemish region)	Organised	Yes	FIT	2013	56–74	2	Yes	69.6		
**Bulgaria**	No programme	NA	NA	NA	NA	NA	NA	NA	10/2014–01/2015	[12,13,20]
**Croatia**	Organised	Yes ^d^	gFOBT	2008	50–74	2	Yes	100.6	04/2014–03/2015	[12,13,16,21]
**Cyprus** (two rural areas of Larnaca)	Organised (pilot)	Yes	FIT	2013	50–69	2	No	No data	09/2014–12/2014	[12,13,16,22]
**Czech Republic**	Opportunistic	Yes	gFOBT	2000–2009	50+	2	NA	NA	06/2014–01/2015	[12,13,16,23,24]
	Opportunistic	Yes	FIT	2009	50–54	1	NA	NA		
	Opportunistic	Yes	FIT/Colonoscopy	2009	55+	2/10	NA	NA		
	Organised	Yes	FIT	2014	50+^e^	2	No	No data		
**Denmark**	Organised	Yes	FIT	2014	50–74	2	Yes	No data	10/2015–12/2015	[12,13,16,25,26]
**Estonia**	Organised (pilot)	Yes	FIT	2016	60–69	2	No	No data	04/2014–12/2014	[12,13,16,27]
**Finland ^f^**	Organised	Yes	gFOBT	2004–2016	60–69	2	Yes	10.5	11/2014–01/2015	[12,13,16,28,29,30]
	Organised	Yes	FIT	2019	60–66	2	Yes	No data		
**France**	Organised	Yes	gFOBT	2002–2015	50–74	2	No	99.1	01/2014–02/2015	[12,13,16,31]
			FIT	2015						
**Germany**	Opportunistic	Yes	gFOBT	1977–2002	45+	1	NA	NA	11/2014–07/2015	[12,32,33]
	Opportunistic	Yes	gFOBT	2002–2017	50–54	1	NA	NA		
			FIT	2017						
	Opportunistic	Yes	gFOBT/colonoscopy	2002–2017	55+	2/10	NA	NA		
			FIT/colonoscopy	2017						
	Organised	Yes	FIT	2019	50–54/55+	1/2	No	NA		
	Organised	Yes	Colonoscopy	2019	50+ (men);55+ (women)	10 (up to 2 colonoscopies)	NA	NA		
**Greece**	Opportunistic	No information	gFOBT	No information	No information	No information	NA	NA	11/2014–03/2015	[12,34,35]
	Opportunistic	No information	Colonoscopy	No information	50–80	5	NA	NA		
**Hungary**	Organised (pilot)	Yes	FIT	2007	50–70	2	No	1.5	10/2014–12/2014	[12,13,16,36]
**Iceland ^g^**	Opportunistic	Yes	Colonoscopy	No information	No information	No information	NA	NA	09/2015–12/2015	[12,37]
**Ireland**	Organised	Yes	FIT	2012	60–69	2	Yes	10.9	10/2014–04/2016	[12,13,16]
**Italy ^h^**	Organised	Yes	gFOBT	1982–1996	50–69	2	No	52.4	10/2015–12/2015	[12,13,16,38,39,40]
			FIT	1996						
**Italy ^i^** (Piedmont and Veneto regions)	Organised	Yes	Sigmoidoscopy	2003/2004	58–60	Once only	NA	No data^j^		
**Latvia**	Opportunistic	Yes	gFOBT	2005	50–74	1	NA	NA	09/2014–02/2015	[12,41]
**Lithuania**	Organised	Yes	FIT	2009	50–74	2	No	No data	09/2014–11/2014	[12,13,16,42]
**Luxembourg**	Opportunistic	Yes	gFOBT/colonoscopy	2005	50+	No information	NA	NA	02/2014–12/2014	[12,13,16]
	Organised	Yes	FIT	2016	55–74	2	Yes	NA		
**Malta**	Organised	Yes	FIT	2013	55–66	2	Yes	28.5	11/2014–08/2015	[12,13,16]
**The Netherlands**	Organised	Yes	FIT	2014	55–75	2	Yes	20.2	01/2014–12/2014	[12,13,16]
**Norway ^k^**	No programme	NA	NA	NA	NA	NA	NA	NA	08/2015–12/2015	[43]
**Poland**	Organised^l^	Yes	Colonoscopy	2012	55–64	Once only	NA	12.5 ^m^	09/2014–12/2014	[12,13,16,44]
**Portugal**	Opportunistic	No information	FIT/Colonoscopy ^n^	No information	50–74	1/10	NA	NA	09/2014–12/2014	[13,16,45,46,47,48]
**Portugal** (Alentejo and Central regions)	Organised (pilot) ^o^	Yes	gFOBT	2009–2018	50–70	2	No information	1.6		
			FIT	2018						
**Romania**	No programme	NA	NA	NA	NA	NA	NA	NA	09/2014–11/2014	[12,13,16,20]
**Slovakia**	Opportunistic	No information	gFOBT/colonoscopy	No information	No information	No information	NA	NA	07/2014–12/2014	[12]
**Slovenia**	Organised	Yes	FIT	2009–2015	50–69	2	Yes	80.0	08/2014–12/2014	[12,13,16,49,50]
				2015	50–74					
**Spain ^p^**	Organised	Yes	gFOBT	2000–2009/2010	50–69	2	Yes	14.2	01/2014–01/2015	[12,13,16,51,52,53]
**Sweden** (Stockholm and Gotland)	Organised	Yes	gFOBT	2008–2015	60–69	2	Yes	8.5	09/2014–01/2015	[12,13,16,54,55]
			FIT	2015						
**UK** (England)	Organised	Yes	gFOBT	2006–2019	60–74 ^q^	2	Yes	54.0	04/2013–03/2015	[12,13,16,56,57]
			FIT	2019						
	Organised	Yes	Sigmoidoscopy	2013	55	Once only	NA	No data		
**UK** (Northern Ireland)	Organised	Yes	gFOBT	2010	60–74	2	Yes	51.0	04/2014–09/2014	[12,13,16,34]
**UK** (Scotland)	Organised	Yes	gFOBT	2007	50–74	2	Yes	110.3	04/2013–03/2015	[12,13,16,58]
			FIT	2017						
**UK** (Wales)	Organised	Yes	gFOBT	2008	60–74	2	Yes	50.1	04/2013–03/2015	[12,13,16,59]
			FIT	2019						

Abbreviations: FIT, faecal immunochemical test; gFOBT, guaiac-based faecal occult blood test; NA, not applicable. ^a^ Information was ordered by (i) country; (ii) type of programme; (iii) year of programme initiation. ^b^ Proportion of invited individuals from the annual target population based on EUROSTAT data from 2013. For Belgium, Malta, the Netherlands and Portugal, the index year was 2014; for Slovenia, the index years were 2011–2012. Data were retrieved from Vale et al. [16]. ^c^ In Austria, gFOBT is undertaken within the freely available annual health check-up for persons aged 40+. ^d^ In Croatia, expenses are reimbursed by health insurance. ^e^ In the Czech Republic, invitations are sent to individuals up to 70 years of age only. ^f^ In Finland, CRC screening was introduced as a randomised, organised health services programme in 2004. By the end of 2012, 22% of the Finnish target population had been invited to a screening. A new programme was initiated in 2019 in nine municipalities (Jyväskylä, Muurame, Orivesi, Oulu, Paimio, Sauvo, Säkylä, Tampere, Ylitornio) and will be gradually expanded to include all 60–74-year olds. ^g^ In Iceland, an organised programme with FIT is foreseen. ^h^ Screening started in 1982 in Florence and in 2000–2004 in other regions. FIT replaced gFOBT in 1996. ^i^ Once only flexible sigmoidoscopy is offered in the Piedmont region and in a small area of the Veneto region. The non-responders are invited to undertake FIT. ^j^ All eligible individuals on the NHS lists are invited for sigmoidoscopy. ^k^ In Norway (counties of Østfold, Akershus and Buskerud), in 2012, about 140,000 individuals aged 50-74 years were invited to participate in a randomised controlled trial to undergo either sigmoidoscopy or FIT. ^l^ In Poland, colonoscopy screening was implemented in the context of randomised health services research and roll-out is ongoing. ^m^ Population aged 55–64 years divided by the number of years of the target age range. The figure was retrieved from Vale et al. [16]. ^n^ In Portugal, despite FIT being the primary screening test recommended by the Ministry of Health, colonoscopy has been found to be the most recommended method by physicians. In 2017, the Portuguese government legislated the implementation of a national organised programme; national extension of the current pilot programme is expected within the next few years. ^p^ In Spain, CRC screening is implemented on a regional level. The programme started in 2000 in Catalonia and has been extended to another eight regions. FIT replaced gFOBT in 2009–2010. ^q^ In August 2018, ministers agreed to further include the younger age groups (50–59 years) in the bowel screening programme, which is now being planned by the National Health Service and Public Health England.

## Data Availability

Data were granted by Eurostat. Information on how to access the data can be found at https://ec.europa.eu/eurostat/web/microdata/european-health-interview-survey.

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
