# Peer review of "Utilisation of Colorectal Cancer Screening Tests in European Countries by Type of Screening Offer: Results from the European Health Interview Survey"

_cancers, 2020, doi:10.3390/cancers12061409_

Round 1

Reviewer 1 Report

Colorectal cancer (CRC) displays a major health burden and metastasis and disease recurrence remain challenging. Thus, early detection of disease is crucial and, indeed, screening methods such as colonoscopy are essential tools to detect CRC in early stages and, consequently, lower mortality rates. In their study, the authors aimed to elucidate the utilization rate of CRC screening methods in European countries. Their findings on this topic indicate the relevance of these tests and will be of interest to the readership of Cancers A number of major and minor points should be addressed prior to publication as detailed below.

Major

To provide a better overview on the benefit of CRC screening methods, the authors should include graphs (looking similar to Fig. 2-4) showing the numbers of CRC-related cases and deaths per 100,000 individuals per country.

Minor

The authors should briefly elaborate on the different types of CRC screening methods (occult blood, stool DNA tests, colonoscopy) and what they are based on.

The first part of the results section (including Figure 1) could be declared as Materials and Methods.

Reviewer 2 Report

Cardoso et al. investigated utilization and types of CRC screening offers in European countries using data from EHIS. The study is potentially interesting, but the authors should address how the difference in utilization and type of CRC screening offers are associated with mortality of CRC.

Reviewer 3 Report

Thank you for the opportunity to review the paper. This paper summarises the similarities and differences of colorectal cancer (CRC) screening programmes in Europe and factors associated with their utilisation. The paper enhances understanding of how CRC screening works across different countries and highlights variations in uptake of interest to stakeholders in those countries.

Abstract:

Line 17 cross sectional

Line 19 potential determinants by type of offer: meaning unclear. Suggest “factors associated with uptake” or other rewording

Line 23: younger age: please include age range in brackets.

Line 25-26: why? Not clear

Line 28-29: I think the implications of the results suggested by the authors do not fit with the main findings and should be reconsidered: while I agree GPs can have some role in signposting to screening, If those who don’t get tests are less likely to attend doctor (longer time since last visit variable) then having physicians as only mechanism to increase utilisation is not likely to reach those who are not presenting

Introduction:

Line 42: “whole population aged…” slightly misleading. Suggest reword to “targeting population aged…”

Line 43 “While some EU countries have meanwhile followed the 43 recommendations and implemented either regional” suggest increase clarity e.g. “some countries have implemented… ,while others …”

Line 52-54: aim of paper does not match the aim in the abstract. I think the paper does 2 things: 1) describes the similarities and differences of crc screening programmes across European countries 2) uses EHIS to look at factors that predict uptake of screening tests.

Results:

Figure 2: y-axis label % of what, please include category; Legend: all EU countries and Iceland and Norway: please see comment in methods and update this description everywhere in manuscript.

Line 90: “proportion of the population up-to-date”: not clear what this means? Please reword.

Line 101+102; report OR and CI in a consistent way indicating what the value is referring to each time.

Please include p value or CI for findings in line 104 and 106.

Line 112 and 113: report CI and OR in consistent way.

Discussion:

I think the relevance of this work in understanding epidemiology and mortality data in CRC should be highlighted in one or two sentences in the discussion: e.g. Documenting the nature and breadth of screening activities across countries should help cross national comparability of screening data, and the development of methodologies to evaluate the mortality impact of population-based screening programmes.

Methods:

Line 215: “survey was conducted in all EU countries, Iceland and Norway” currently 27 members in EU, UK not. Please describe number of countries included here and how selected.

Line 225: please add reference

Suggest table 1 is included in results section. Think this would help reader understand rest of the comparisons.

Round 2

Reviewer 1 Report

All previous concerns have been addressed by the authors.

Reviewer 2 Report

The authors have adequately addressed the issue I raised.